# Infodemiology of RSV in Italy (2017–2022): An Alternative Option for the Surveillance of Incident Cases in Pediatric Age?

**DOI:** 10.3390/children9121984

**Published:** 2022-12-16

**Authors:** Matteo Riccò, Antonio Baldassarre, Sandro Provenzano, Silvia Corrado, Milena Pia Cerviere, Salvatore Parisi, Federico Marchesi, Marco Bottazzoli

**Affiliations:** 1AUSL–IRCCS di Reggio Emilia, Servizio di Prevenzione e Sicurezza Negli Ambienti di Lavoro (SPSAL), Local Health Unit of Reggio Emilia, 42122 Reggio Emilia, Italy; 2Department of Experimental and Clinical Medicine, University of Florence, 50134 Florence, Italy; 3Local Health Unit of Trapani, ASP Trapani, 91100 Trapani, Italy; 4Department of Medicine DAME—Division of Pediatrics, University of Udine, 33100 Udine, Italy; 5Università Cattolica del Sacro Cuore, 00168 Rome, Italy; 6Sanofi, Medical Affairs, 20100 Milan, Italy; 7Department of Medicine and Surgery, University of Parma, 43126 Parma, Italy; 8Department of Otorhinolaryngology, APSS Trento, 31223 Trento, Italy

**Keywords:** RSV, respiratory syndromes, lower respiratory tract infection, infodemiology

## Abstract

The aim of this study was to evaluate whether or not online queries for Respiratory Syncytial Virus (RSV) retrieved by means of Google Trends™ and the Italian Wikipedia analysis program mirror the occurrence of influenza-like illnesses (ILI), as reported by the Italian Influenza Surveillance network (InfluNet). Estimated rates for ILI in the general population and in the age groups 0–4 years and 5–14 years were obtained for the influenza seasons 2017–2018 to 2020–2021. Similarly, a weekly fraction of online searches was retrieved for a series of terms associated with Respiratory Syncytial Virus. Next, trends for daily visualization of Italian Wikipedia Pages for Human Respiratory Syncytial Virus, Pneumonia, Bronchiolitis, Influenza, and Respiratory Failure were similarly retrieved. The correlation of all search terms with ILI was analyzed by means of Spearman’s rank correlation analysis. Among search terms associated with the clinical diagnosis of Respiratory Syncytial Virus infections, the occurrence of ILI was highly correlated only with Bronchiolitis in the age group 0–4 years (β 0.210, *p* = 0.028), while more generic search terms, such as Bronchitis, fever, influenza, and Pneumonia, were identified as effective predictors of ILI, in general and by age groups. In a regression analysis modeled with ILIs as the outcome variable, daily visualizations for the Wikipedia pages on Bronchiolitis were identified as negative predictors for ILI in general (β = −0.152, *p* = 0.032), ILI in age group 0–4 years (β = −0.264, *p* = 0.001) and 5–14 years (β = −0.202, *p* = 0.006), while Influenza was characterized as a positive effector for ILIs in the age group 5–14 years (β = 0.245, *p* = 0.001). Interestingly, not only were the search terms extensively correlated with one another, but all of them were also characterized by autocorrelation through a Durbin-Watson test (all estimates DW < 2.0) In summary, our study identified a complicated pattern of data visualization as no clear association between rates of ILI in pediatric age group 0–4 and 5 to 14 years was actually found. Finally, our data stress that the infodemiology option may be quite problematic for assessing the time trend of RSV infections in Italy until more appropriate reporting will be made available, by sharing estimates of Lower Respiratory Tract Infections, and through a more accurate characterization of younger age groups.

## 1. Introduction

Respiratory Syncytial Virus (RSV) is a common cause of Influenza-Like Illnesses (ILI) and lower-respiratory tract infections (LRTIs), particularly among newborns and infants [1,2,3,4]. RSV-associated LRTIs are predominantly characterized as Bronchiolitis or Pneumonia [5], accounting for around 60% to 80% of infant Bronchiolitis and up to 40% of pediatric pneumonias [6,7]. While the rate of hospitalization due to RSV has been conservatively estimated, allegedly, it ranges, worldwide, between 1.2% [8] and 1.6% [9]; being among the five primary diagnoses in hospitalized infants < 1 year of age [9,10,11,12,13], greater uncertainty affects the estimates on the total burden of disease. According to some recent estimates, before the inception of the SARS-CoV-2 pandemic, a total of 33–35 million cases occurred annually in children younger than five years of age, with a clearly defined and somewhat predictable seasonal epidemic [9,14,15,16]. More precisely, in the Northern Hemisphere (including the USA, the UK, France, Germany, and Italy), RSV outbreaks usually begin in the calendar months of November or December, reaching the annual by the end of January/early February, and ending by March or April [17,18], thus overlapping with seasonal influenza epidemics [19]. Nonetheless, as the large majority of incident cases of RSV infections are managed as outpatients, without a proper microbiological diagnosis, the actual number of incident cases remains largely unknown [1,4]. Even though World Health Organization has started a global effort for developing reliable and international standards for RSV surveillance, RSV surveillance in the EU is fragmentary, only involving 20 out of 27 member states, and quite heterogeneous in terms of the collected data [20]. Notably, neither official data on RSV infections nor LRTIs for Italy had been made available until the 2022 reporting season [19,21].

Since 2000, the occurrence of ILI in Italy has been monitored by the Italian Influenza Surveillance Network (InfluNet), which combines epidemiological and virological surveillance in order to track influenza epidemics, as well as the spatio-temporal spread and circulation of respiratory pathogens during the “influenza season” (i.e., from mid-October to the late April of the subsequent year) [22,23,24]. The InfluNet relies on the collaboration between regional departments of health and the voluntary participation of sentinel physicians (general practitioners and pediatricians), who survey approximately 2% of the general population (increased to 4% since the 2020–2021 influenza season), ensuring the representativeness of all age groups (0–4 years, 5–14 years, 15–64 years, and ≥65 years, respectively), with homogeneous geographical distribution [24]. Although the InfluNet was upgraded in 2020 in order to track the occurrence of RSV infections, until the reporting season 2022–2023, reports from the InfluNet did not provide data on RSV circulation [4,19], and actual figures on the seasonal circulation of RSV have remained largely uncertain.

While some observational studies based on the regional database have provided some snapshots of the clinical data [4,19,21,25,26,27,28,29], alternative options for monitoring RSV infection trends are therefore required, at least until epidemiological and virological surveillance is able to deliver appropriate and reliable information. In recent years, infodemiology (i.e., the science of distribution and determinants of information in an electronic medium, specifically the Internet, or in a population, with the ultimate aim to inform public health and public policy) [30] and infoveillance (i.e., epidemic surveillance performed by means of infodemiology) [31,32] have emerged as effective tools in predicting outbreaks of several infectious diseases, ranging from influenza to COVID-19 [32,33,34,35,36,37,38,39,40]. The rationale of infodemiology is that the appropriate analysis of research trends in specific search engines, web platforms and social media may reflect or even anticipate the epidemiological features of certain disorders [41,42,43,44,45,46,47]. As suggested by the pioneering study of Eysenbach in 2006, Internet searches may even anticipate doctors’ visits to sentinel physicians by around one week, as often people first consult the Internet (colloquially, “Doctor Google”) before going to the doctor [30,40,43].

In the case of RSV, internet searches may be of particular interest for patients of pediatric age, as their parents—most of whom lack a personal medical background—may rely on internet information sources to find explanations and/or solutions for their child’s clinical conditions or to retrieve information on otherwise uncommon diagnoses, such as Bronchiolitis, and even the very same term “Respiratory Syncytial Virus”.

In this study, we therefore aimed to assess whether a correlation exists between internet search volumes for items associated with RSV and a potential proxy for RSV diagnoses, i.e., official figures of InfluNet programs on ILI, specifically focusing on figures from pediatric age groups (0–4 and 5–14 years). In terms of information sources, we focused on the Google search engine and Wikipedia. To the best of our knowledge, there is no study investigating the relationships between Google Trends^TM^ and Wikipedia searches on RSV and the occurrence of Influenza-Like Syndromes.

## 2. Materials and Methods

### 2.1. Epidemiological Data

Since the winter season of 2000, the epidemiological and virological surveillance plan for ILI (i.e., InfluNet) has implemented the publication of weekly reports during the “influenza season” (i.e., week 42 to week 17 of the subsequent calendar year) [4,22,23,24,48], and correspondent estimates are freely available on the portal of the Italian National Institute of Health dedicated to the Integrated Influenza Surveillance System (https://w3.iss.it/site/rmi/influnet/Default.aspx; accessed on 19 October 2022). According to the European Center for Disease Prevention and Control’s (ECDC) case definition of ILI [49], InfluNet epidemiological surveillance includes cases characterized by: abrupt onset of fever (>38°), one or more respiratory symptoms (non-productive cough, sore throat, rhinitis), and one or more systemic symptoms (myalgia, headache and severe malaise). In infants and children (i.e., age group 0–4 years, and 5–4 years, respectively), manifestations of ILI also include: abrupt onset of high fever, coryza, cough, sore throat, vomiting and/or diarrhea (particularly in breastfeeding infants), abdominal pain, fatigue, headache, red eye, conjunctivitis, and myalgia [22,23,24,50,51]. From the available weekly reports, we retrieved weekly incidence rates for ILI for the total population, and by age groups: 0–4, 5–14,. All of the data were reported at the national level.

### 2.2. Internet Search Volumes

In Italy, Google is the main search engine, and a significant share of the Italian population utilizes it for searching information on common medical problems, including communicable and non-communicable diseases [32,44]. Since 2006, Google^TM^ (now Alphabet Inc., Mountain View, CA, USA) has developed an internal app (Google Trends^TM^) aimed to analyze the popularity of search queries in Google Search across various region and languages, being repeatedly implemented as a reliable information source for infodemiology studies [30,40,45,46,47].

On the other hand, Wikipedia is a multilingual free online encyclopedia that is among the largest and most-read reference works in history [41,42]. Italian Wikipedia has similarly emerged as a highly accessed and referenced information source, having specifically designed, web-based apps that implements analysis of users’ queries on Wikipedia itself.

The search strategy is outlined in Table 1. More precisely, the data on Google Research Volumes were retrieved by means of Google Trends^TM^ (Alphabet Inc, Mountain View, CA, USA; https://trends.google.it/; accessed on 19 October 2022), the web-based applet developed by Google that shows how frequently a given search term is entered into Google’s search engine within a certain timeframe. Research volumes are reported as 0 to 100 values, relative to the site’s total search volume over a given period of time (i.e., Relative Search Volumes). We only considered queries from Italy input between 16 October 2017 and 16 October 2022, on a combination of search terms that, according to the recommendation of the United States CDC for the general population, may deal with potential clinical diagnoses of LRTIs (i.e., “bronchiolite”, eng. Bronchiolitis; “Virus Respiratorio Sinciziale”, eng. Respiratory Syncytial Virus; “VRS”, eng. RSV; “bronchite”, eng. Bronchitis;, “polmonite”, “eng. Pneumonia; insufficienza respiratoria”, eng. Respiratory Failure; “raffreddore”, eng. Common Cold; “influenza”, eng. Influenza/Flu), and a series of symptoms that have been specifically associated with RSV infections, i.e., “febbre” (eng. fever), “tosse” (eng. cough), “starnuti” (eng. sneezing), “respire sibilante” (eng. wheezing), “rinorrea” (eng. runny nose), “inappetenza” (eng. decrease in appetite) [52].

No combinations of words were included in the analyses, while only one term per search was eventually assessed. In contrast to Google Trends^TM^, the Wikipedia application programming interface easily allows users to retrieve data on page visualizations rather than on search queries. Moreover, the geographical origin of the web searches is not available for alleged privacy and security reasons. As a consequence, we retrieved, through the web applet Pageviews Analysis^TM^ (https://pageviews.wmcloud.org; accessed on 19 October 2022), data on daily visualizations of a series of pages from the Italian Wikipedia on the following topics: “bronchiolite” (eng. Bronchiolitis), “Virus Respiratorio Sinciziale Umano” (eng. Human Respiratory Syncytial Virus), “bronchite” (eng. Bronchitis), “polmonite” (eng. Pneumonia), “influenza” (eng. Influenza). The Italian language is almost exclusively spoken and used in Italy, and is the only language used by the overwhelming majority of the Italian population; therefore, it is very unlikely that cumulative figures of Italian Wikipedia web searches may have been substantially influenced by queries performed abroad and, at the same time, it is debatable that a large share of Italian people would carry out searches in other languages. All searches were performed before 20 October 2022.

### 2.3. Statistical Analyses

We initially performed descriptive analysis of the surveillance data on ILI by calculating the estimates for each reporting seasons. In order to underline and then track down the impact of the SARS-CoV-2 pandemic on the transmission of respiratory pathogens, including RSV [53,54,55], we assumed the reporting seasons of 2017–2020—during which time no preventive measures were actually taken—as a reference and calculated the estimates for the excess incidence rates (EIR) in 2020–2021, i.e., the season that was most heavily affected by lockdown and non-pharmaceutical interventions (NPI), and 2021–2022, when confinement measures and NPI were substantially lifted. NPI have been defined as interventions that communities can take to help slow the spread of illnesses that aim to prevent and/or control the pathogen’s transmission in the community) [6,7]. For the purposes of this study, EIR was defined as the difference between the reported incidence rates [RIR] in a given week in 2020–2021 and 2021–2022 and the estimate of the expected incidence rate [EXR], calculated as an average for the index week i in the assessed timeframe, for the time period 2017–2020, as follows:EIR = (RIR^i^ − EXR^i,2017–2020^)/EXR^i,2017–2020^
where:

RIR^i,a^ = reported rates in a given week i

EXR^i,2017–2020^ = average rates in the given week i for the time period 2017–2020

Comparisons between the seasonal incidence rates were then performed by means of Analysis of the Variance (ANOVA), with a Dunnet’s post-hoc test that assumed the reporting season 2017–2018 as the reference group.

Corresponding estimates for Relative Search Volumes on Google Research Terms, and for the average daily visualization were calculated for the corresponding calendar weeks of the reporting seasons: 2017–2018, 2018–2019, 2019–2020, 2020–2021, and 2021–2022. Similarly to the estimates for ILI, the data for the timeframe 2017–2020 were assumed as a reference, and the difference in the estimates for the reporting seasons 2020–2021 and 2021–2022 were calculated accordingly.

The average estimates for Relative Search Volumes and Wikipedia page visualization during the reporting seasons (i.e., from week 42 to week 17 of the subsequent calendar year; “in season”) and during the warm season (i.e., from week 18 to week 42 of the calendar year; “out of season”) were then calculated accordingly and compared by means of a Mann-Whitney U test.

The relationships between the estimates for ILI by age groups and the infodemiology data from Google Trends^TM^ and Italian Wikipedia were initially investigated through the calculation of the Spearman’s rank correlation coefficient (ρ). All research terms that were significantly correlated (i.e., *p* < 0.05) with the ILI estimates in the univariate analysis were included as explanatory variables in the regression analysis models, in which the ILI estimates were the related outcome variables (model 1: total population; model 2: age group 0–4 years; model 3: age group 5–14).

In order to assess whether or not the assessed search volumes were correlated with one another, a correlation that included all of the infodemiological variables was then calculated by means of the Spearman’s rank correlation coefficient.

Finally, the infodemiological variables were assessed in terms of their potential autocorrelation. Autocorrelation can be defined as the correlation of a certain factor with itself over time, suggesting the underlying cyclic pattern of reported data [32]. Potential autocorrelation was ascertained through the calculation of the Durbin-Watson (DW) statistics. The DW test is a statistic test used to detect the presence of autocorrelation in the residuals (prediction errors) from a regression analysis [56]. The DW test statistic or d always lies between 0 and 4. If the d is substantially less than 2, there is evidence of positive serial correlation, while values greater than 2 suggest no autocorrelation.

All of the calculations were performed on R 4.0.3 (R Core Team (2020). R: A language and environment for statistical computing. R Foundation for Statistical Computing, Vienna, Austria. URL https://www.R-project.org/; accessed on 11 November 2022)) [57] by means of packages acnr (v. 1.0.0), EpiReport (v 1.0.1), fmsb (0.7.0), R.utils (2.12.0).

### 2.4. Ethical Approval

No ethical approval was needed for this study, as no individual data were identifiable and we only analyzed and presented aggregated data.

## 3. Results

### 3.1. Incidence of ILI in Italy

The occurrence of ILI during the assessed timeframe (i.e., 16 October 2017–16 October 2022) is graphically reported in Figure 1, in general and by age groups. Interestingly, since 2017–2018 to 2019–2020, the seasonal peak was consistently identified in the early weeks of January (weeks 1 to 4 for 2017–2018; weeks 3 to 6 in 2018–2019; weeks 4 to 7 in 2019–2020). During the winter season of 2020–2021, the incidence rates were substantially lower than those reported in previous years, and during the following season, 2021–2022, a distinctive time trend was visually recognizable. More precisely, the seasonal peak for the reporting season 2020–2021 shifted from January to February at lower incidence rates. On the other hand, despite a certain recovery compared to the reporting season 2020–2021, the following winter season 2021–2022 was characterized by two distinctive peaks, in the early and late stages of the reporting seasons (i.e., during the months of December 2021 and April 2022).

As shown in Table 2, the average incidence rate for ILI was 5.11 per 1000 persons (95%CI 3.29 to 6.94) in 2017–2018, decreasing to 1.44 (95%CI 1.32 to 1.56) in 2020–2021 (*p* < 0.001 compared to the reporting season 2017–2018), with a resurgence to 3.94 (95%CI 3.61 to 4.27) in 2021–2021. A substantially decreased EIR was then estimated to −69.94% (95%CI −89.26 to −50.62) for 2020–2021, while a slightly decreased EIR in 2021–2022 (−17.85%, 95%CI −64.10 to +28.39). A similar trend was identified in all age groups, with a substantial decrease in incidence rates for 2020–2021 (all age groups, *p* < 0.001 compared to the reporting season 2017–2018). More precisely, there was a respective drop in the incidence rate in pediatric age groups (i.e., 0–4 years, and 5 to 14 years) was −78.89% (95%CI −91.06 to −66.72), and −82.38% (−99.40 to −65.36) for reporting season 2020–2021 compared to 2017–2020. Whilst the average estimates for 2021–2022 were somehow lower than those reported in 2017–2020, the difference was not substantial (i.e., −8.31%, 95%CI −88.71 to +72.10 for 0–4 years; −32.66%, 95%CI −99.68 to +34.36 for 5–14 years).

### 3.2. RSV-Related Queries

Relative search volumes of selected keywords in each week from 16 October 2017 to 16 October 2022 from Google Trends^TM^ are reported in Appendix B, Figure A1 and Figure A2, while Figure A3 reports the daily visualization estimates for Italian Wikipedia pages that were included in the analyses.

As shown in Table 3, during the assessed timeframe, the greater search volumes were associated with cough (45.04, 95%CI 42.18 to 47.90), followed by Bronchitis (39.91, 95%CI 36.91, 42.91), fever (31.11, 95%CI 29.14, 33.08), decrease in appetite (30.83, 95%CI 29.13, 32.54), respiratory failure (30.22, 95%CI 27.70 to 32.74). Search terms such as Respiratory Syncytial Virus, RSV and Bronchiolitis (3.84, 95%CI 2.03 to 5.65; 3.76, 95%CI 2.37 to 5.15; 14.5, 95%CI 11.80 to 17.31, respectively) were characterized by low or very low research volumes. Focusing on the Wikipedia daily visualizations, the greatest figures were associated with pages on Pneumonia (638.82, 95%CI 516 to 761.49), Influenza (602, 95%CI 402.33 to 801.70), Bronchitis (192.44, 95%CI 165.68 to 219.20), Respiratory Syncytial Virus (88.67, 955CI 47.69 to 129.65), and eventually Bronchiolitis (67.57, 95%CI 56.42 to 78.73).

Comparisons of research estimates is reported in Table 4.

Some distinctive trends were identified. Assuming the reporting season 2017–2018 as a reference group, the search volumes of Bronchitis and Bronchiolitis substantially decreased in 2019–2020 and in 2020–2021, while estimates for 2021–2022 were not substantially increased. On the contrary, the research volumes for Respiratory Syncytial Virus were substantially increased in 2021–2022 compared to the reference season.

Search volumes for respiratory symptoms such as sneezing and runny nose increased from the reference year 2017–2018 (18.69, 95%CI 16.76 to 20.63, and 10.46, 95%CI 8.85 to 12.08) to substantially increased estimates in 2019–2020 (32.33, 95%CI 28.35 to 36.31, and 17.65, 955CI 14.88; 20.42; *p* < 0.001 for both comparisons), 2020–2021 (33.00, 95%CI 30.58 to 35.42, and 25.35, 95%CI 23.57 to 27.03, *p* < 0.001 for both comparisons), and 2021–2022 (36.33 95%CI 32.31 to 40.34 and 31.19, 95%CI 27.20 to 35.18, *p* < 0.001 for both comparisons). Interestingly, research terms such as Pneumonia (16.92, 95%CI 11.57 to 22.28, *p* = 0.001 when compared to the reference season), and respiratory failure (33.37, 95%CI 27.87 to 38.86) peaked during the reporting season 2019–2020; however, the latter estimate was not significantly increased compared to the baseline values for 2017–2018 (*p* = 0.087).

Subsequently, search terms such as fever (34.48, 95%CI 32.12 to 36.84), cough (54.02, 95%CI 50.06 to 57.98), and common cold (35.42, 95%CI 30.34 to 40.51) were associated with significantly increased research volumes in 2021–2022 when compared to 2017–2018 (in all cases, *p* < 0.001), with influenza also reporting an increased research volume for 2019–2020 (19.56, 95%CI 13.50; 25.61).

Similarly, the Wikipedia daily visualizations for Influenza peaked in 2019–2020 compared to the reference year (1147.31 visualizations/day vs. 254.89, *p* < 0.001). On the contrary, the rates for Bronchitis were higher in the reference year 2017–2018 (335.18 visualizations/day) than in all of the following reporting seasons (182.05 in 2018–2019; 137.36 in 2019–2020; 45.10 in 2020–2021, and 34.97 in 2021–2022). When dealing with searches on the Wikipedia page for Pneumonia, estimates for 2017–2018, 2018–2019 and 2019–2020 were quite similar (668.89 in 2017–2018, compared to 597.85 in 2018–2019, and 872.26 in 2019–2020), with a substantial decrease in 2020–2021 (298.54, *p* = 0.002) and in 2021–2022 (148.09, *p* < 0.001).

The EIR estimates for search terms and Wikipedia pages are shown in Table 5, while comparisons of the search volumes before and after the presumptive inception of the pandemic (i.e., December 2019) are provided as Appendix A. Whilst the research volumes for Bronchiolitis substantially decreased in 2020–2021 compared to 2017–2020 (−74.50%, 95%CI −82.83 to −66.16), they in turn increased by +255.62% (95%CI +35.02 to +478.21) in 2021–2022, with a similar habit for Respiratory Syncytial Virus (−51.73%, 95%CI −94.73 to −8.73 in 2020–2021, and +356.16%, 95%CI +4.48 to +708.85 in 2021–2022). On the contrary, web searches for Bronchitis substantially decreased in 2020–2021 compared to the reference timeframe 2017–2020 (−61.17%, 95%CI −68.11 to −54.23), with a resurgence to previous volumes in 2021–2022. When focusing on the main symptoms, search volumes for cough were characterized by a biphasic trend, with a decrease in 2020–2021 compared to 2017–2020 (−38.38%, 95%CI −49.20 to −27.56), and a substantial resurgence in 2021–2022 (+35.43%, 95%CI +18.41 to +52.44), while sneezing and runny nose were characterized by a substantial increase in volume searches both in 2020–2021 (+52.93%, 95%CI +30.11 to +75.75 and +102.94%, 95%CI 72.35 to 133.54) and in 2021–2022 (+82.97%, 95%CI +48.34 to +117.60, and +212.27%, 95%CI +158.80 to +265.73). Interestingly, searches for common cold and decrease in appetite were only increased in 2021–2022 (+81.08%, 95%CI +57.36 to 104.80, and +16.33%, 95%CI +5.75 to +26.90), while search volumes for influenza were substantially decreased in both 2020–2021 (−30.16%, 95%CI −55.25 to −5.07) and 2021–2022 (−24.67%, 95%CI −40.39 to −8.96) compared to 2017–2020.

No substantial differences were identified for RSV, Pneumonia, and wheezing, in both 2020–2021 and 2021–2022 compared to the baseline timeframe, 2017–2020.

Regarding the daily visualization rates for Wikipedia, the estimates for bronchiolitis substantially decreased in 2020–2021 compared to the previous timeframe (−80.77%, 95%CI −84.34 to −77.20). Similarly to the visualizations of the page for Human Respiratory Syncytial Virus (−31.40%, 95%CI −52.13 to −10.74), the following reporting season was associated with daily rates that were otherwise in line with the reference years 2017–2020 (−20.68%, 95%CI −92.26 to +50.90, and +418.81%, 95%CI −425.28 to +1263.91, respectively). Eventually, the daily visualization rates for Wikipedia pages for Influenza (−48.63%, 95%CI −88.19 to −9.08 in 2020–2021; −75.86%, 95%CI −97.21 to −54.51 in 2021–2022), Bronchitis (−81.87%, 95%CI −83.85 to −79.75; and −83.82%, 95%CI −85.84 to −81.79), and Pneumonia (−54.92%, 95%CI −67.43 to −43.42, and −80.05%, 95%CI −84.69 to −75.40), were consistently reduced in 2020–2021 and 2021–2022 compared to 2017–2020.

The aforementioned search items were compared in terms of the inquiries during the reporting season and during the summer season (Table 6). In fact, all of the research terms and Wikipedia pages were characterized by higher research and visualization volumes during the winter season when compared to the summer season, with the notable exceptions of RSV (3.76, 95%CI 2.37 to 5.15 “in season” vs. 3.14, 95%CI 2.96 to 3.32 “out of season”, *p* = 0.220), and running nose (19.96, 95%CI 17.66 to 22.27 vs. 17.75, 95%CI 13.57 to 18.71, *p* = 0.434).

### 3.3. Correlation between Research Terms and Estimates for ILI

The correlation estimates between the ILI in general and by age groups (0–4 years, 5–14 years) are provided in Table 7. The ILI calculated for the whole of the assessed population was positively correlated with search volumes for Bronchiolitis (ρ = 0.726, *p* < 0.001), Respiratory Syncytial Virus (ρ= 0.185, *p* = 0.029), Bronchitis (ρ= 0.826, *p* < 0.001), Pneumonia (ρ= 0.290, *p* = 0.001), fever (ρ= 0.605, *p* < 0.001), common cold (ρ = 0.686, *p* < 0.001), decrease in appetite (ρ = 0.228, *p* = 0.007), and influenza (ρ = 0.688, *p* < 0.001). Similar correlations were found also in age groups 0–4 and 5–14 years, with the notable exception of Pneumonia in age group 0–4 years (ρ = 0.103, *p* = 0.224).

Regarding the correlation between the ILI and Italian Wikipedia web page visualizations, a positive correlation was consistently identified across the various age groups for Bronchiolitis, Human Respiratory Syncytial Virus, and Bronchitis. On the contrary, in both of the assessed age groups (i.e., 0–4 and 5–14 years), no actual correlation was found between the ILI rates and daily visualizations of Pneumonia (ρ = 0.072, *p* = 0.400, and ρ = 0.112, *p* = 0.186, respectively), and the age group 0–4 also did not correlate with the visualizations for Bronchitis (ρ = 0.132, *p* = 0.120).

The correlation of research terms and daily visualizations for Wikipedia pages is reported in full detail in Appendix B. When focusing on research terms associated with main diagnoses (i.e., RSV, Respiratory Syncytial Virus, and Bronchiolitis), a substantial correlation was reported only for Respiratory Syncytial Virus and Bronchiolitis (ρ = 0.262, *p* < 0.001; Figure A4). On the contrary, the research terms associated with main symptoms and daily visualizations for Wikipedia pages associated with main diagnoses were extensively correlated with one another (Table A1, Table A2, Table A3 and Table A4).

### 3.4. Regression Analysis

Therefore, the regression analysis models were modelled as follows (Appendix A):Model 1 (Total population) included the ILI rates as outcome variables, and assessed the search volumes for brochiolitis, Respiratory Syncytial virus, Bronchitis, Pneumonia, fever, decrease in appetite, cough, influenza, and all of the assessed web searches from Wikipedia as explanatory variables.Model 2 (age 0–4 years) included all of the aforementioned explanatory variables, with the exception of the search volumes Pneumonia; similarly, the estimates for the Wikipedia visualizations for influenza and Pneumonia were excluded.Model 3 (age 5–14 years) included all of the explanatory variables included in model 1 with the exception of the daily visualization rates for Pneumonia.

A significant regression equation was found for all of the aforementioned models (Appendix A). More precisely (Table 8), the estimates for ILI found a significant and positive predictor in internet search volumes for Bronchitis for all of the considered age groups (standardized coefficient β = 0.588, *p* < 0.001 for total ILI; β = 0.983, *p* < 0.001 for ILI 0–4 years; β = 0.563, *p* = 0.001 for ILI 5–14 years), while Bronchiolitis was characterized as a positive predictor in the sole subgroup ILI 0–4 years (β = 0.210, *p* = 0.028). Moreover, internet searches for Pneumonia were identified as a negative predictor in both models in which it was included (total ILI: β = −0.560, *p* = 0.002, ILI 5–14 years: β = −0.740, *p* < 0.001).

Interestingly, fever and influenza were characterized as positive predictors for ILI in total (β = 0.424, *p* = 0.002 for fever; β = 0.410, *p* = 0.002 for influenza) and the in age group 5–14 years (β = 0.322, *p* = 0.009 for fever; β = 0.529, *p* < 0.001 for influenza), while no substantial effect was associated with ILI in infants aged 0–4 years. On the contrary, the search volumes for common cold were characterized as a negative predictor for total ILI (β = −0.309, *p* = 0.002), as well as for ILI 5–14 years (β = −0.310, *p* = 0.005), while no effect was identified in the age group 0–4 years.

Finally, the daily visualizations for Wikipedia pages on Bronchiolitis were identified as negative predictors for ILI in general (β = −0.152, *p* = 0.032), as well as ILI in the age groups 0–4 years (β = −0.264, *p* = 0.001) and 5–14 years (β = −0.202, *p* = 0.006), while Influenza was characterized as a positive effector for the age group 5–14 years (β = 0.245, *p* = 0.001).

### 3.5. Time Series

The estimates for autocorrelation in the search terms are summarized in Annex Figure A5 and Figure A6. The Durbin-Watson statistics were characterized by estimates that were <2.0 in all of the analyses, not only for the volume search terms from Google Trends^TM^ (Annex Figure A5), but also when dealing with inquiries from Italian Wikipedia (Annex Figure A6). In other words, a cyclic, seasonal trend was eventually identified for all of the studied search terms.

## 4. Discussion

RSV has increasingly emerged as a major pathogen, particularly in infants and children [58], but also in older adults [59,60]. As the actual nation-wide burden of disease for RSV in Italy still remains only vaguely defined, in this study, we assessed whether or not infodemiology may represent an alternative option for monitoring its epidemiological trend. In a multivariable regression model, the research volumes for Bronchiolitis were eventually characterized among the main effectors for reported ILI, but only in the age group 0–4 years. In other words, an increased research volume for this term associated with the main clinical diagnosis of RSV infection could be acknowledged as somehow predictive of ILI in young infants. To our knowledge, this was the first study investigating whether Google Trends^TM^ and Wikipedia searches on RSV and related search terms may represent a reliable proxy for the actual time trend of RSV infections in Italy. The referral to this innovative approach for tracking infectious disease has been characterized as both effective and reliable, but no previous study on RSV had been previously performed [31,32,38,39,43]; in fact, our research has reported some contradictory results, thus requiring further studies.

For example, RSV has been affected by the SARS-CoV-2 pandemic in a particular and somehow unexpected way. Whilst evolutionarily unrelated, SARS-CoV-2 and RSV shares several epidemiological features [1], and NPI aimed to counter the spreading of the former have also been quite effective against RSV, the reporting of which experienced an abrupt end prompted by the implementation of non-pharmaceutical interventions in the 2020 season [53,54,55,61,62,63,64]. According to available estimates, the occurrence of RSV infections has substantially dropped in Italy [4,21]; however, following the lifting of NPI and lockdown measures, subsequent reports have stressed an unprecedented resurgence of RSV circulation among susceptible infants [25,27,28,65]. From this point of view, research volumes for keywords such as Respiratory Syncytial Virus, its acronym RSV, and Bronchiolitis, did experience a sudden and substantial surge during the last week of 2021, with similar features from the analysis of the daily visualization of Wikipedia web pages. The retrieved trends mirrored some regional estimates on the incidence of RSV cases [66,67] and even multicentric reports [66]: compared to the pre-pandemic era, admission for Bronchiolitis sharply decreased during the first year of the SARS-CoV-2 pandemic (i.e., −87% during 2020–2021), with a substantial resurgence during the 2021–2022 (i.e., +369% compared to 2020–2021), and 1177% compared to 2021–2022. In addition, our data were affected by a similar seesaw trend, seemingly suggesting a consistent correlation with the actual circulation of the pathogen.

Some precautionary remarks are still needed. Firstly, not only the research terms and Wikipedia web pages dealing with RSV experienced such a trend, but also those associated with other respiratory tract infections and their features (i.e., cough, −38.38%, 95%CI −49.20; −27.56 in 2020–2021 compared to 2017–2020, and +35.43% (+18.41; +52.44) in 2021–2022). Moreover, we noticed that the search volumes for influenza remained significantly below the reference timeframe 2017–2020, not only in 2020–2021 (−30.16%, 95%CI −55.25 to −5.07), but also in 2021–2022 (−24.67%, 95%CI −40.39 to −8.96), with similar features from the analyses of the Wikipedia visualization trends (−48.63%, 95%CI −88.19 to −9.08 in 2020–2021, and −75.86%, 95%CI −97.21 to −54.51 in 2021–2022). Despite the fact that the search volumes and seasonal ILI estimates for 2021–2022 shared the biphasic trend, with two distinctive incidence peaks, the global estimates were comparable to the reference years 2017–2022, both in the general population and in all of the assessed age groups. A potential explanation may be found in the very same research volumes for RSV. The keyword RSV (Italian “VRS”) was quite rarely inquired by Google users during all of the assessed timeframes, with the notable exception of the peak reported during November 2021. While the rationale behind the infodemiological research would suggest that these findings may mirror the actual emergence of this disorder and, therefore, the real-world circulation of the pathogen, a more reasonable approach would point towards the sharing of this acronym among potential stakeholders who, deprived of a specific medical background, simply researched required information through the Internet [40,41,42]. In other words, the unprecedented surge of RSV during 2021–2022 has reclaimed the interest of both medical research and traditional media [3,19,24,68]. Interestingly, some previous reports from Italian parents have stressed a limited knowledge of this pathogen [2,69]. Therefore, we cannot rule out that these research peaks may be the indirect consequence of the higher interest of the general population towards something otherwise perceived as uncommon and unfamiliar, rather than representing a proxy of the RSV circulation among the targeted population.

Another precautionary remark should be placed on the outcome variable of the nationwide estimates for ILI, as the World Health Organization has recently stressed that ILI may represent an improper proxy for RSV infections. In fact, severe acute respiratory illnesses (ARI) and LRTI definitions, which also include Bronchiolitis and Pneumonia, would represent a far better proxy for this pathogen, but again the correspondent seasonal data on the Italian population remains either fragmentary or unavailable [4,48]. Therefore, the lack of the actual and consistent predictive effect on research volumes for RSV and Respiratory Syncytial Virus can be explained through the inconsistency and substantial imbalance between what Public Health professionals would need to measure when dealing with RSV, and what available surveillance data actually report [19,29,65,66,70]. Not coincidentally, in the regression analysis models, search terms such as Influenza and Bronchitis were identified as significant predictors for ILI, while the visualization rates for the Wikipedia page on Bronchiolitis were characterized as negative predictors for ILI in pediatric age groups of infants (0–4 years) and children (5–14 years). In this regard, another substantial shortcoming is represented by the age groups that have been defined and made available from the InfluNet reports [19,22,24].

From a pediatric point of view, the age of group 0–4 years encompasses infants that are quite heterogeneous both in terms of risk factors and needs [1,4,11,68], particularly when dealing with respiratory infectious diseases [71,72,73,74]. Focusing on RSV, epidemiological studies have stressed how maternal antibodies reduce the risk of RSV infection in infants during the first months of extra-uterine life. Therefore, while the risk of RSV infections is usually very low in the first month of extrauterine life, around half of children are infected within their first year of life, and the chance of having developed an RSV-related infection reaches almost 100% by the age of 2 [4,75,76,77,78,79,80,81] or 3 years of age [81,82]. In the SARS-CoV-2 pandemic setting, the extensive implementation of NPI and lockdown measures in Italy, as well as in the rest of European Union, has dramatically increased the number of infants who did not develop any immunity against RSV, which now not only includes infants older than 1 month, but also potentially involving infants aged 3 to 4 years [3,83]. As a consequence, while the main targets for RSV infections in the timeframe 2017–2020 should be identified in infants aged between 1–2 months and 2 years, during reporting seasons 2020–2021 and 2021–2022, we should expect an unprecedented involvement of older infants, up to 3–4 years of age. Unfortunately, the available data impair a more accurate analysis.

*Limits*. In addition to the aforementioned critical issue represented by the limited representativity of ILI for actual RSV cases, our study is affected by several other limits that should be accurately addressed. First and foremost, although the InfluNet surveillance network has been designed in order to provide estimates that could be acknowledged as representative of the general Italian population [19,22,24,51,84], its original design specifically targeted influenza and influenza-like syndromes, and it can deliberately miss LRTIs because of its clinical features, which are far more representative of RSV than ILI themselves [1,2,20,29,85]. In this regard, it should be stressed that while the European data have identified a certain circulation of RSV at the national level, even after the end of conventional surveillance for ILI, Italy did not extend the surveillance season in 2020–2021 or in 2021–2022, compromising the capability of the InfluNet to properly track down RSV infections [58,86,87].

Second, the surveillance systems primarily target Seasonal and Pandemic Influenza, the spreading of which is usually characterized by nationwide and even larger epidemics, including pandemics. Moreover, as the referral to Italian primary care professionals is mostly paid through general taxation, the potential under-reporting of ILI, particularly in certain socio-economic groups, could be acknowledged as relatively low [19,22,24]. On the contrary, enforced surveillance systems may fail to properly track a pathogen such as RSV, the circulation of which is primarily based on local, low-rate transmission [88,89,90,91]. In this regard, while seasonal and pandemic influenza guarantee some sort of long-term immunity against their respective pathogens, as unexpectedly stressed by the recent A(H1N1)v pandemic [50,51,92,93], RSV infections elicit a short-lived mucosal immunity that reduces the risk for systemic complications in the subsequent months, but it does not abolish the risk for new, symptomatic infections, as stressed by the epidemic of 2021–2022 [1,2,53,54,55,61,62,63,85].

Another substantial limitation to be addressed is represented by the reduced scalability of our data from a geographical point of view. While Google Trends^TM^ guarantees a georeferentiation for all queries, sub-national and regional estimates are more scarcely reliable and accurate. Moreover, the Wikimedia API deliberately impairs users’ ability to retrieve page visualizations by geographical area: officially, the scope of this policy is to avoid any sort of discrimination towards the users of this platform, but clearly it affects the potential representativity of the collected results. Nonetheless, as the Italian language is unfrequently spoken outside of Italy, at least compared to English, French, Spanish, or German, it is quite unlikely that the collected data could have been altered by international users. Conversely, as a large share of the Italian general population has substantial difficulties in handling any foreign language (>40% by 2017 estimates) [94], it is reasonable that a substantial share of the Italian general population did prefer localized Wikipedia (i.e., Italian) pages over the international website.

Third, the first stages of the SARS-CoV-2 pandemic and the related lockdowns were associated with a sustained increase in the use of internet services, particularly where broadband connections were available. Whilst the overall status of internet connections is usually acknowledged as far from optimal, an extensive use of internet services was identified in Italy [95,96]. As a consequence, we cannot rule out that the increased research volumes identified during the reporting seasons 2019–2020, particularly in its late weeks, and then in 2020–2021, may have been affected by the unprecedent referral to the new media for retrieving required information, with further limits when comparing the post-pandemic figures with the pre-pandemic ones.

Fourth, several factors included in the analyses were significantly correlated with one another, as shown in full details in the Appendix B. In other words, we cannot rule out that some degree of collinearity among the different variables that were included in the regression analysis models may have impaired their actual reliability. This is particularly important when taking into consideration that respiratory infectious diseases are quite difficult to discriminate between each other [20,29,58].

Fifth, it should be stressed that the early stages of the SARS-CoV-2 pandemic mostly occurred unnoticed to the Italian general population: as recently pointed out by some studies [97,98,99,100], well before the official diagnosis of first cases of COVID-19 (i.e., 21 February 2020), Italy reasonably experienced a certain circulation of this pathogen. The clinical characteristics of milder cases may have therefore led to a transient surge in both searches for topics correlated with respiratory disorders and to ILI notification rates, that then were conversely affected by the implementation of lockdown measures [99,101,102,103]. Nevertheless, despite intensive retrospective research, particularly on a series of viral Pneumonia cases that occurred in the second half of 2019, the actual circulation of SARS-CoV-2 before February 2020 remains unclear [104,105,106,107]. Thus, our analysis reports a frankly subtitled trend for 2020, a bias likely linked to the impact of the pandemic still underway. The SARS-CoV-2 pandemic has, in fact, entailed a series of cascading side effects that have impacted public health, limiting access to primary care and, indirectly, leading to an under-reporting of health and disease states distinct from COVID-19.

Finally, the most significant caveat resulted from the calculation of the autocorrelation plots and from the analysis of the reported variables by means of the Durbin-Watson test. In fact, all of the variables that were included in the analyses exhibited some sort of autocorrelation (See Appendix B). Autocorrelation is sometimes known as serial correlation in the discrete time case, i.e., the correlation of a signal with a delayed copy of itself as a function of delay [32,33,34]. A high degree of autocorrelation, such as in our study, suggests that the data reflect an underlying cyclic pattern of events that actually drives the results. In our study, search terms for respiratory signs and symptoms and for Respiratory Syncytial Virus (i.e., Bronchiolitis, Respiratory Syncytial Virus, and RSV), exhibited a substantial degree of autocorrelation; therefore, we could speculate that seasonality represented their main driver, with environmental and behavioral features as the root causes of the monitored trends.

## 5. Conclusions

The infodemiological analysis has already been possible for some decades thanks to the computing power of IT tools, but it was definitively consecrated with the advent of the SARS-CoV-2 pandemic. At present, it represents an innovative option for monitoring infectious diseases, either as a compliment for more conventional approaches (i.e., virological and epidemiological surveillance) or as a “buffer” substitution when other options are limitedly available or entirely unavailable. In fact, our analysis suggested a limited correspondence between the research volumes for symptoms and the web searches associated with RSV at national level and some estimates by Italian sentinel physicians included in the InfluNet Surveillance system. Interestingly, the web searches for the term Bronchiolitis were characterized as significant effectors for ILI estimates in the 0–4 age group. Unfortunately, as this age group is quite heterogenous, and as the definition of ILI is limitedly consistent with the working definition of RSV infection, until a proper re-design of the InfluNet is made available by also including LRTI cases and a more accurate distinction between pediatric age groups, no further analyses could be performed, and even the present estimates should be handled with care and a precautionary approach.

## Figures and Tables

**Figure 1 children-09-01984-f001:**
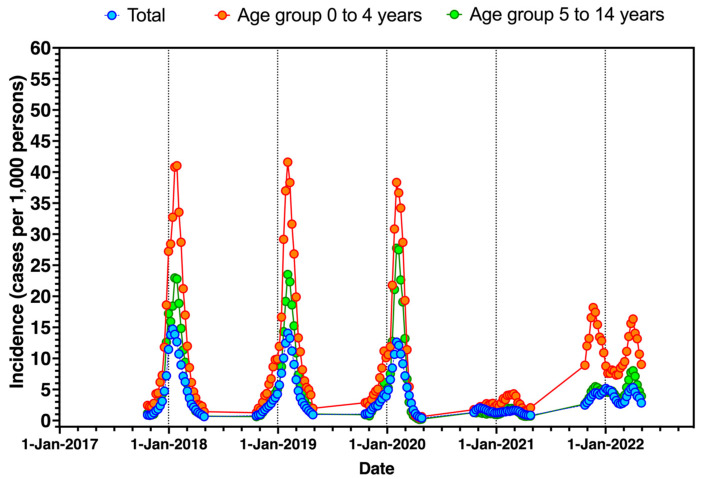
Seasonal trend of influenza-like illnesses (ILI) as reported by InfluNet reports (2017–2022), in general and by pediatric age groups (i.e., 0 to 4 and 5 to 14 years). Dotted lines represent 31 December of plotted calendar years.

**Table 1 children-09-01984-t001:** Checklist for analysis of Google Research Volumes and Wikipedia Visualizations.

Section/Topic	Google Research Volume	Wikipedia
Access Date	20 October 2022	20 October 2022
Time period	16 October 2017–16 October 2022	16 October 2017–16 October 2022
Search Tool	Google Trends^TM^ (https://trends.google.it/)	Pageviews Analysis (https://pageviews.wmcloud.org/)
Limits	Searches from Italy	Only pages in Italian
Search Input (Italian)	Bronchiolitis (“bronchiolite”) Respiratory Syncytial Virus (“Virus Respiratorio Sinciziale”) RSV (“VRS”) Bronchitis (“bronchite”) Pneumonia (“polmonite”) common cold (“raffreddore”) influenza (“influenza”) respiratory failure (“insufficienza respiratoria”) fever (“febbre”) Cough (“tosse”) wheezing (“respire sibilante”) sneezing (“starnuti”) runny nose (“naso che cola”) decrease in appetite (“inappetenza”)	Bronchiolitis (“bronchiolite”) Human Respiratory Syncytial Virus (“Virus Respiratorio Sinciziale Umano”) Bronchitis (“bronchite”) Pneumonia (“polmonite”) influenza/flu (“influenza”)
Combination	Only one search term	Only one search term
Retrieved data	Relative Search Volumes	Number of Daily Visualizations

**Table 2 children-09-01984-t002:** Incidence rates (cases per 1000 persons) for influenza-like illnesses (ILI; InfluNet surveillance system of the Italian National Health Institute), and corresponding excess incidence rates calculated for 2020–2021 and 2021–2022 compared to the averages for 2017–2020. Comparisons between seasonal incidence rates were performed by means of analysis of the variance (ANOVA) with post-hoc test of Dunnet, assuming reporting season 2017–2018 as the reference one.

	Seasonal Incidence Rate (Cases per 1000 Persons; 95%CI)					Excess Incidence Rates (%, 95%CI)	
	2017–2018	2018–2019	2019–2020	2020–2021	2021–2022	2017–2020 vs. 2020–2021	2017–2020 vs. 2021–2022
0–4 years	13.46 (8.65; 18.26) [REFERENCE]	13.06 (8.52; 17.60) [*p* = 1.000]	11.37 (6.94; 15.80) [*p* = 0.839]	2.67 (2.35; 2.98) [*p* < 0.001]	11.58 (10.32; 12.84) [*p* = 0.882]	−78.89% (−91.06.; −66.72)	−8.31% (−88.71; +72.10)
5–14 years	7.41 (4.64; 10.18) [REFERENCE]	6.91 (4.32; 9.49) [*p* = 0.994]	7.11 (3.90; 10.31) [*p* = 0.999]	1.26 (1.12; 1.40) [*p* = 0.001]	4.81 (4.32; 5.30) [*p* = 0.307]	−82.38% (−99.40; −65.36)	−32.66% (−99.68; +34.36)
TOTAL	5.11 (3.29; 6.94) [REFERENCE]	4.79 (3.19; 6.38) [*p* = 0.988]	4.49 (3.04; 5.95) [*p* = 0.888]	1.44 (1.32; 1.56) [*p* < 0.001]	3.94 (3.61; 4.27) [*p* = 0.475]	−69.94% (−89.26; −50.62)	−17.85% (−64.10; +28.39)

**Table 3 children-09-01984-t003:** Relative Search volumes of selected terms (overall estimate 2017–2021).

Relative Search Volumes	Average, 95% Confidence Interval
Bronchiolitis	14.56 (11.80; 17.31)
RSV	3.76 (2.37; 5.15)
Respiratory Syncytial Virus	3.84 (2.03; 5.65)
Bronchitis	39.91 (36.91; 42.91)
Pneumonia	13.26 (11.15; 15.36)
Fever	31.11 (29.14; 33.08)
Cough	45.04 (42.18; 47.90)
Sneezing	28.99 (26.57; 31.40)
Wheezing	12.11 (9.66; 14.56)
Decrease in appetite	30.83 (29.13; 32.54)
Respiratory Failure	30.22 (27.70; 32.74)
Runny Nose	19.96 (17.66; 22.27)
Common Cold	27.74 (25.55; 29.92)
Influenza	16.14 (13.57; 18.71)
**Wikipedia Daily Visualizations**	**Average, 95% Confidence Interval.**
Bronchiolitis	67.57 (56.42; 78.73)
Influenza	602.17 (402.33; 801.70)
Respiratory Syncytial Virus	88.67 (47.69; 129.65)
Bronchitis	192.44 (165.68; 219.20)
Pneumonia	638.82 (516.16; 761.49)

**Table 4 children-09-01984-t004:** Relative search volumes for the assessed research terms, average and correspondent 95% confidence intervals (95%CI). Comparisons were performed by mean of Analysis of the Variance (ANOVA) with Dunnet’s post hoc test (reference category [REF]: reporting season 2017–2018).

	Relative Search Volumes (Average, 95%CI.)				
	2017–2018	2018–2019	2019–2020	2020–2021	2021–2022
Bronchiolitis	9.52 (7.05; 11.99) [REF.]	10.19 (7.82; 12.56) [*p* = 0.996]	7.77 (5.39; 10.15) [*p* = 0.043]	3.28 (2.48; 4.08) [*p* = 0.012]	16.92 (10.22; 23.62) [*p* = 0.999]
RSV	3.33 (3.00; 3.66) [REF.]	3.10 (3.00; 3.66) [*p* = 0.999]	2.69 (2.44; 2.95) [*p* = 0.957]	2.91 (2.66; 3.15) [*p* = 0.990]	5.35 (1.58; 9.11) [*p* = 0.269]
Respiratory Syncytial Virus	1.65 (0.73; 2.58) [REF.]	1.85 (0.90; 2.80) [*p* = 1.000]	1.63 (0.82; 2.45) [*p* = 1.000]	1.47 (0.59; 2.34) [*p* = 1.000]	6.44 (1.73; 11.16) [*p* = 0.010]
Bronchitis	37.31 (32.47; 42.14) [REF.]	36.79 (32.29; 41.29) [*p* = 0.999]	29.37 (23.65; 35.08) [*p* = 0.028]	17.72 (16.13; 19.30) [*p* < 0.001]	38.25 (35.07; 41.43) [*p* = 0.993]
Pneumonia	7.58 (6.27; 8.88) [REF.]	8.08 (7.43; 8.72) [*p* = 0.996]	16.92 (11.57; 22.28) [*p* < 0.001]	10.51 (9.06; 11.95) [*p* = 0.316]	9.31 (8.05; 10.57) [*p* = 0.754]
Fever	24.88 (22.47; 27.29) [REF.]	23.96 (22.11; 25.82) [*p* = 0.964]	32.33 (28.29; 36.36) [*p* < 0.001]	29.09 (27.41; 30.78) [*p* = 0.073]	34.48 (32.12; 36.84) [*p* < 0.001]
Cough	34.77 (30.59; 38.95) [REF.]	35.06 (30.95; 39.17) [*p* = 1.000]	35.85 (29.90; 41.79) [*p* = 0.990]	27.72 (25.19; 30.24) [*p* = 0.065]	54.02 (50.06; 57.98) [*p* < 0.001]
Sneezing	18.69 (16.76; 20.63) [REF.]	19.33 (17.46; 21.20) [*p* = 0.995]	32.33 (28.35; 36.31) [*p* < 0.001]	33.00 (30.58; 35.42) [*p* < 0.001]	36.33 (32.31; 40.34) [*p* < 0.001]
Wheezing	10.94 (7.36; 14.53) [REF.]	14.08 (8.75; 14.53) [*p* = 0.681]	12.44 (7.16; 17.73) [*p* = 0.965]	9.64 (6.60; 12.69) [*p* = 0.979]	13.94 (10.40; 17.49) [*p* = 0.712]
Decrease in appetite	31.67 (29.29; 34.06) [REF.]	26.92 (24.53; 39.32) [*p* = 0.045]	30.00 (26.59; 33.41) [*p* = 0.792]	31.87 (29.50; 34.23) [*p* = 1.000]	32.25 (32.44; 38.05) [*p* = 0.681]
Respiratory Failure	27.33 (23.35; 31.30) [REF.]	26.00 (23.07; 28.93) [*p* = 0.967]	33.37 (27.87; 38.86) [*p* = 0.087]	25.49 (22.15; 28.83) [*p* = 0.901]	25.42 (22.60; 28.25) [*p* = 0.891]
Runny Nose	10.46 (8.85; 12.08) [REF.]	9.96 (8.53; 11.39) [*p* = 0.996]	17.65 (14.88; 20.42) [*p* < 0.001]	25.30 (23.57; 27.03) [*p* < 0.001]	31.19 (27.20; 35.18) [*p* < 0.001]
Common Cold	19.90 (17.41; 22.40) [REF.]	19.77 (17.42; 22.12) [*p* = 1.000]	22.06 (18.49; 25.63) [*p* = 0.781]	24.64 (21.85; 27.43) [*p* = 0.152]	35.42 (30.34; 40.51) [*p* < 0.001]
Influenza	10.94 (7.57; 14.31) [REF.]	9.50 (7.44; 11.56) [*p* = 0.928]	19.56 (13.50; 25.61) [*p* = 0.001]	7.30 (6.21; 8.39) [*p* = 0.331]	8.65 (7.42; 9.89) [*p* = 0.724]
	**Wikipedia Daily visualizations (Average, 95%CI)**				
Bronchiolitis	53.16 (46.90; 59.41) [REF.]	79.96 (59.42; 100.50) [*p* = 0.037]	41.30 (32.69; 49.91) [*p* = 0.614]	22.15 (13.43; 30.86) [*p* = 0.011]	42.39 (19.28; 65.50) [*p* = 0.700]
Influenza	254.89 (206.23; 301.95) [REF.]	209.78 (182.92; 236.63) [*p* = 0.996]	1147.31 (636.49; 1568.13) [*p* < 0.001]	301.19 (250.26; 352.13) [*p* = 0.995]	138.08 (103.87; 172.29) [*p* = 0.894]
Respiratory Syncytial Virus	26.81 (23.22; 30.40) [REF.]	26.04 (22.51; 29.57) [*p* = 1.000]	57.29 (44.89; 69.68) [*p* = 0.943]	126.66 (3.19; 250.12) [*p* = 0.181]	158.90 (42.51; 275.28) [*p* = 0.050]
Bronchitis	335.18 (297.66; 382.69) [REF.]	182.05 (155.75; 208.34) [*p* < 0.001]	137.36 (103.43; 171.28) [*p* < 0.001]	45.10 (41.27; 48.92) [*p* < 0.001]	34.97 (30.04; 39.91) [*p* < 0.001]
Pneumonia	668.89 (609.23; 728.55) [REF.]	597.85 (518.76; 676.94) [*p* = 0.906]	872.26 (556.07; 1188.45) [*p* = 0.170]	298.54 (259.96; 337.12) [*p* = 0.002]	148.09 (135.76; 160.42) [*p* < 0.001]

**Table 5 children-09-01984-t005:** Excess volume searches for research terms in Google Trends and in Wikipedia (Italy). In the analyses, average estimates for 2017–2019 were assumed as the reference categories.

Research Field	Excess Volume Search Rates (%, 95%CI)	
	2017–2020 vs. 2020–2021	2017–2020 vs. 2021–2022
Google Trends^TM^		
Bronchiolitis	−74.50% (−82.83; −66.16)	+255.62% (+35.02; +478.21)
RSV	−4.45% (−15.89; +24.78)	+118.18% (−26.16; +262.52)
Respiratory Syncytial Virus	−51.73% (−94.73; −8.73)	+356.16% (+4.48; +708.85)
Bronchitis	−61.17% (−68.11; −54.23)	+3.89 (−13.32; +21.11)
Pneumonia	+37.37% (−2.00; +76.75)	+1.78% (−19.50; +23.06)
Fever	+8.67% (−11.99; +29.33)	+17.36% (+6.89; +27.84)
Cough	−38.38% (−49.20; −27.56)	+35.43% (+18.41; +52.44)
Sneezing	+52.93% (+30.11; +75.75)	+82.97% (+48.34; +117.60)
Wheezing	+40.50% (−35.59; +116.59)	+55.53% (−25.95; +138.02)
Decrease in appetite	+0.21% (−10.77; +11.20)	+16.33% (+5.75; +26.90)
Respiratory Failure	−1.35% (−19.23; +16.52)	−16.68% (−34.66; +1.30)
Running Nose	+102.94% (+72.35; +133.54)	+212.27% (+158.80; +265.73)
Common Cold	−8.44% (−23.42; +6.54)	+81.08% (+57.36; +104.80)
Influenza	−30.16% (−55.25; −5.07)	−24.67% (−40.39; −8.96)
Wikipedia		
Bronchiolitis	−80.77% (−84.34;−77.20)	−20.68% (−92.26; +50.90)
Influenza	−48.63% (−88.19; −9.08)	−75.86% (−97.21; −54.51)
Respiratory Syncytial Virus	−31.40% (−52.13; −10.74)	+418.81% (−425.28 +1263.91)
Bronchitis	−81.87% (−83.85; −79.75)	−83.82% (−85.84; −81.79)
Pneumonia	−54.92% (−67.43; −43.42)	−80.05% (−84.69; −75.40)

**Table 6 children-09-01984-t006:** Comparisons of the average relative search volumes for Google Trends^TM^ keywords and daily visualizations from Italian Wikipedia for the reporting season (i.e., October–April; “in season”) vs. non reporting season (i.e., May to September; “out of season”). Data were reported with their correspondent 95% confidence intervals (95%CI). Analyses were performed by means of the Mann-Whitney test (M-W).

Research Field	In Season (Average, 95%CI)	Out of Season (Average, 95%CI)	M-W U (*p* Value)
Google Trends^TM^	Relative Search Volumes		
Bronchiolitis	14.56 (11.80; 17.31)	3.68 (3.14; 4.22)	13,474.5, *p* < 0.001
RSV	3.76 (2.37; 5.15)	3.14 (2.96; 3.32)	7758.0, *p* = 0.220
Respiratory Syncytial Virus	3.84 (2.03; 5.65)	1.17 (0.67; 1.68)	10,184.0, *p* < 0.001
Bronchitis	39.91 (36.91; 42.91)	22.49 (20.86; 24.12)	13,432.0, *p* < 0.001
Pneumonia	13.26 (11.15; 15.36)	7.26 (6.67; 7.86)	13,852.0, *p* < 0.001
Fever	31.11 (29.14; 33.08)	26.45 (25.18; 27.73)	10,552.5, *p* = 0.001
Cough	45.04 (42.18; 47.90)	28.65 (26.19; 31.12)	13,260.5, *p* < 0.001
Sneezing	28.99 (26.57; 31.40)	26.76 (24.76; 28.76)	8942.5, *p* = 0.437
Wheezing	12.11 (9.66; 14.56)	12.30 (9.40; 15.19)	8612.5, *p* = 0.806
Decrease in appetite	30.83 (29.93; 32.37)	31.50 (29.74; 33.27)	8279.5, *p* = 0.754
Respiratory Failure	30.22 (27.70; 32.74)	24.38 (22.21; 26.55)	10,500.0, *p* = 0.001
Running Nose	19.96 (17.66; 22.27)	17.75 (13.57; 18.71)	8946.0, *p* = 0.434
Common Cold	27.74 (25.55; 29.92)	20.45 (18.13; 22.78)	11,756.0, *p* < 0.001
Influenza	16.14 (13.57; 18.71)	5.43 (4.97; 5.89)	15,342, *p* < 0.001
Wikipedia	Daily Visualizations		
Bronchiolitis	67.57 (56.42; 78.73)	24.24 (19.81; 28.67)	12,709.5, *p* < 0.001
Influenza	88.67 (47.70; 129.65)	66.21 (10.85; 121.56)	12,527.0, *p* < 0.001
Respiratory Syncytial Virus	192.44 (165.68; 219.20)	94.92 (77.54; 112.31)	12,009.0, *p* < 0.001
Bronchitis	638.82 (516.16; 761.49)	380.28 (323.21; 437.35)	11,044.5, *p* < 0.001
Pneumonia	602.02 (402.34; 801.70)	188.37 (161.37; 215.37)	14,068.0, *p* < 0.001

**Table 7 children-09-01984-t007:** Correlation between weekly notification rates for ILI and relative volume searches for a series of Google Trends^TM^ keywords, and daily visualizations for selected Italian Wikipedia pages (2017 to 2022).

	Influenza-like Illnesses (2017–2022)		
Research Field	0–4 Years	5–14 Years	TOTAL
Google Trends^TM^			
Bronchiolitis	ρ = 0.776 *p* < 0.001	ρ = 0.735 *p* < 0.001	ρ = 0.726 *p* < 0.001
RSV	ρ = 0.111 *p* = 0.193	ρ = 0.042 *p* = 0.620	ρ = 0.035 *p* = 0.679
Respiratory Syncytial Virus	ρ = 0.206 *p* = 0.015	ρ = 0.172 *p* = 0.042	ρ = 0.185 *p* = 0.029
Bronchitis	ρ = 0.839 *p* < 0.001	ρ = 0.827 *p* < 0.001	ρ = 0.826 *p* < 0.001
Pneumonia	ρ = 0.103 *p* = 0.224	ρ = 0.169 *p* = 0.046	ρ = 0.290 *p* = 0.001
Fever	ρ = 0.456 *p* < 0.001	ρ = 0.524 *p* < 0.001	ρ = 0.605 *p* < 0.001
Cough	ρ = 0.769 *p* < 0.001	ρ = 0.777 *p* < 0.001	ρ = 0.812 *p* < 0.001
Sneezing	ρ = −0.043 *p* = 0.614	ρ = 0.020 *p* = 0.811	ρ = −0.075 *p* = 0.380
Wheezing	ρ = 0.016 *p* = 0.847	ρ = −0.007 *p* = 0.932	ρ = 0.037 *p* = 0.668
Decrease in appetite	ρ = 0.247 *p* = 0.003	ρ = 0.239 *p* = 0.005	ρ = 0.228 *p* = 0.007
Respiratory Failure	ρ = −0.024 *p* = 0.777	ρ = 0.021 *p* = 0.810	ρ = 0.051 *p* = 0.547
Running Nose	ρ = 0.034 *p* = 0.692	ρ = 0.038 *p* = 0.652	ρ = 0.088 *p* = 0.300
Common Cold	ρ = 0.680 *p* < 0.001	ρ = 0.689 *p* < 0.001	ρ = 0.686 *p* < 0.001
Influenza	ρ = 0.553 *p* < 0.001	ρ = 0.569 *p* < 0.001	ρ = 0.688 *p* < 0.001
Wikipedia			
Bronchiolitis	ρ = 0.429 *p* < 0.001	ρ = 0.395 *p* < 0.001	ρ = 0.365 *p* < 0.001
Influenza	ρ = 0.416 *p* < 0.001	ρ = 0.413 *p* < 0.001	ρ = 0.500 *p* < 0.001
Respiratory Syncytial Virus	ρ = 0.275 *p* = 0.001	ρ = 0.395 *p* < 0.001	ρ = 0.283 *p* = 0.001
Bronchitis	ρ = 0.132 *p* = 0.120	ρ = 0.204 *p* = 0.016	ρ = 0.241 *p* = 0.004
Pneumonia	ρ = 0.072 *p* = 0.400	ρ = 0.112 *p* = 0.186	ρ = 0.214 *p* = 0.011

**Table 8 children-09-01984-t008:** Regression analysis for the outcome variables represented by Influenza-Like Illnesses (ILI) estimates and main assessed keywords from Google Trends^TM^, and topics from Italian Wikipedia, that in univariate analysis were associated by Spearman’s test *p* < 0.001. Analyses were performed by total sample (i.e., Total), and age groups 0–4 years, and 5–14 years.

Variable	ILI Age Group	B	95%CI		β	t	*p* Value
Bronchiolitis	TOTAL	0.030	−0.006	0.066	0.141	1.654	0.101
	0–4 yrs	0.133	0.015	0.251	0.210	2.223	0.028
	5–14 yrs	0.057	−0.139	0.057	0.147	1.688	0.094
Respiratory Syncytial Virus	TOTAL	−0.028	−0.080	0.023	−0.088	−1.095	0.276
	0–4 yrs	−0.094	−0.273	0.085	−0.098	−1.043	0.299
	5–14 yrs	−0.041	−0.139	0.057	−0.070	−0.831	0.407
Bronchitis	TOTAL	0.115	0.051	0.178	0.588	3.586	<0.001
	0–4 yrs	0.568	0.404	0.732	0.983	6.857	<0.001
	5–14 yrs	0.200	0.080	0.320	0.563	3.294	0.001
Pneumonia	TOTAL	−0.155	−0.250	−0.060	−0.560	−3.241	0.002
	0–4 yrs	-	-	-	-	-	-
	5–14 yrs	−0.374	−0.493	−0.255	−0.740	−6.226	<0.001
Fever	TOTAL	0.122	0.047	0.196	0.410	3.235	0.002
	0–4 yrs	0.056	−0.160	0.272	0.063	0.511	0.610
	5–14 yrs	0.174	0.066	0.304	0.322	2.638	0.009
Cough	TOTAL	−0.004	−0.076	0.068	−0.020	−0.110	0.913
	0–4 yrs	−0.208	−0.440	0.024	−0.342	−1.771	0.079
	5–14 yrs	−0.016	−0.155	0.122	−0.044	−0.236	0.814
Influenza	TOTAL	0.096	0.035	0.158	0.424	3.104	0.002
	0–4 yrs	0.038	−0.142	0.218	0.056	0.419	0.676
	5–14 yrs	0.219	0.104	0.335	0.529	3.753	<0.001
Common Cold	TOTAL	−0.081	−0.134	−0.029	−0.305	−3.061	0.003
	0–4 yrs	−0.056	−0.224	0.112	−0.071	−0.664	0.508
	5–14 yrs	−0.151	−0.251	−0.050	−0.310	−2.966	0.005
Decrease in appetite	TOTAL	0.017	−0.019	0.053	0.050	0.941	0.348
	0–4 yrs	0.109	−0.010	0.228	0.107	1.813	0.072
	5–14 yrs	0.040	−0.026	0.106	0.063	1.190	0.236
WIKI–Human Respiratory Syncytial Virus	TOTAL	0.001	−0.001	0.003	0.062	0.753	0.453
	0–4 yrs	0.005	−0.004	0.013	0.108	1.106	0.271
	5–14 yrs	0.002	−0.003	0.006	0.062	0.715	0.476
WIKI–Bronchiolitis	TOTAL	−0.008	−0.015	−0.001	−0.152	−2.173	0.032
	0–4 yrs	−0.041	−0.066	−0.016	−0.264	−3.262	0.001
	5–14 yrs	−0.019	−0.033	−0.006	−0.202	−2.782	0.006
WIKI–Influenza	TOTAL	0.001	0.000	0.001	0.198	1.494	0.138
	0–4 yrs	0.001	−0.001	0.002	0.066	0.885	0.378
	5–14 yrs	0.001	0.001	0.002	0.245	3.531	0.001
WIKI–Pneumonia	TOTAL	−0.001	−0.003	0.001	−0.164	−0.692	0.490
	0–4 yrs	-	-	-	-	-	-
	5–14 yrs	-	-	-	-	-	-
WIKI–Bronchitis	TOTAL	0.000	−0.004	0.004	−0.005	−0.054	0.957
	0–4 yrs	-	-	-	-	-	-
	5–14 yrs	−0.004	−0.010	0.002	−0.100	−1.368	0.174

Note: B = unstandardized regression coefficient; β = standardized regression coefficient; 95%CI = 95% confidence interval.

## Data Availability

Raw data are available from the Study Authors.

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
