# Peer review of "Infodemiology of RSV in Italy (2017–2022): An Alternative Option for the Surveillance of Incident Cases in Pediatric Age?"

_children, 2022, doi:10.3390/children9121984_

Round 1

Reviewer 1 Report

Overall this is a very interesting and well presented study.  There are modest grammatical issues such as line 56 (perhaps uncertainty is better term), line 58 (occurred annually), line 109 (does not do) and lines 154 and 155 (not complete sentence--words missing???).  Abstract line 36--missing word Fever and ???--line 416 suggests it should be Influenza??

The idea is an important one to study.  Interestingly the figures suggest better correlation than do the analyses and this may be confusing.  Figure 1 is a very nice display of data.

The results are very number dense and slow reading making the tables and figures very important.

While the study covers a long period, the noted issues with the SARS2 pandemic suggest that post hoc analyses that separate the study into periods prior to December 2019 and after that might be more descriptive of predictive abilities.

Lines 260--the early October RSV in 2021 seems to be predictive of early 2022 RSV in US???

Figure 3---the late 2019 surge may be predictive of the pandemic recognized in early 2020 as suggested by authors in lines 655-665 which might be stressed further.

Line 574--the media is mentioned by were there nightly new stories about RSV that would have triggered this---perhaps more indepth report here.

Line 608---after the pandemic were 3 to 4 yo really at low risk or still increased risk due to the low periods during pandemic.

The conclusions are of course disappointing to some but excellent suggestions are made regarding needs for better data inputs.

Author Response

Estimated Rev. 1,

We thank you for your valuable comments. In fact, we substantially agreed with all of your recommendations, that were therefore implemented in the main text. And more precisely:

Overall this is a very interesting and well presented study.  There are modest grammatical issues such as line 56 (perhaps uncertainty is better term), line 58 (occurred annually), line 109 (does not do) and lines 154 and 155 (not complete sentence--words missing???).  Abstract line 36--missing word Fever and ???--line 416 suggests it should be Influenza??

Thank you. Not only the shortcomings addressed by Rev.1 have been addressed, but the Abstract has been substantially rewritten.

The idea is an important one to study.  Interestingly the figures suggest better correlation than do the analyses and this may be confusing.  Figure 1 is a very nice display of data.

Thank you. In fact, according to the recommendations of Rev.2 we’ve also simplified Figure 1.

The results are very number dense and slow reading making the tables and figures very important.

While the study covers a long period, the noted issues with the SARS2 pandemic suggest that post hoc analyses that separate the study into periods prior to December 2019 and after that might be more descriptive of predictive abilities.

Thank you, we simplified the data reporting, and some of them have moved to the Supplementary Appendix A and B. We think that this choice would improve the understanding of our estimates.

Regarding the requirement for dichotomizing estimates in before and after December 2019, we included a post-hoc analysis as Supplementary Table S1.

Lines 260--the early October RSV in 2021 seems to be predictive of early 2022 RSV in US???

We discussed this topic across the text, but deliberately included a cautious assessment for reasons that are now included in the discussion among the limits:

Third, the first stages of SARS-CoV-2 pandemic and the associated lockdowns were associated with a sustained increase in the use of internet services, particularly where broadband connections were available. Even though the overall status of internet con-nections is usually acknowledged as far from optimal, an extensive use of internet services was identified also in Italy [95,96]. As a consequence, we cannot rule out that the increased research volumes identified during the reporting seasons 2019-2020, particularly in its late weeks, and then 2020-2021 may have been affected by the unprecedent referral to the new media for retrieving the needed information, with further limits when comparing post-pandemic figures with pre-pandemic ones.

Fifth, it should be stressed that the early stages of the SARS-CoV-2 pandemic mostly occurred unnoticed to the Italian general population: as recently pointed out by some studies [97–100], well before the official diagnosis of first cases of COVID-19 (i.e. 21 February 2020), Italy reasonably experienced a certain circulation of this pathogen. Clinical characteristics of milder cases may have therefore led to a transient surge in both searches for topics correlated with respiratory disorders and to ILI notification rates, that then were conversely affected by the implementation of lockdown measures [99,101–103]. Still, despite an intensive retrospective research, particularly on a series of viral pneumonia cases that occurred in the second half of 2019, the actual circulation of SARS-CoV-2 before February 2020 still remains unclear [104–107]. Nevertheless, our analysis reports a frankly subtitled trend for 2020, a bias likely linked to the impact of the pandemic still underway. The SARS-CoV-2 pandemic has, in fact, entailed a series of cascading side effects that impacted public health, limiting access to primary care and, indirectly, leading to an under-reporting of health and disease states distinct from COVID-19.

Figure 3---the late 2019 surge may be predictive of the pandemic recognized in early 2020 as suggested by authors in lines 655-665 which might be stressed further.

Line 574--the media is mentioned by were there nightly new stories about RSV that would have triggered this---perhaps more indepth report here.

We totally agree with this point, and we therefore included the following section:

While the rationale behind the infodemiological research would suggest that these findings may mirror the actual emergence of this disorder and therefore the real-world circulation of the pathogen, a more reasonable approach would point towards the sharing of this acronym among potential stakeholders who, deprived of specific medical back-ground, simply did research needed information through the Internet [40–42]. In other words, the unprecedented surge of RSV during 2021-2022 has reclaimed the interest not only of medical research, but also by traditional media [3,19,24,68]. Interestingly, some previous reports from Italian parents have stressed a limited knowledge of this pathogen [2,69]. Therefore, we cannot rule out that these research peaks may be the indirect con-sequence of the higher interest of the general population towards something otherwise perceived as uncommon and unfamiliar, rather than representing a proxy of the RSV circulation among the targeted population.

Line 608---after the pandemic were 3 to 4 yo really at low risk or still increased risk due to the low periods during pandemic.

We again agree with Rev.1 and included the following section in the discussion:

In the SARS-CoV-2 pandemic settings, the extensive implementation of NPI and lockdown measures in Italy as well as in the rest of European Union has dramatically increased the number of infants who did not develop any immunity against RSV, that now not only includes infants older than 1 month, potentially involving infants aged 3 to 4 years [3,83]. As a consequence, while main targets for RSV infections in the timeframe 2017-2020 should be identified in infants aged 1-2 months to 2 years, during reporting seasons 2020-2021 and mostly 2021-2022 we should expect an unprecedented involvement of older infants, up to 3-4 years of age. Unfortunately, available data impair a more accurate analysis.

Again, thank you for the time you’ve spent in the analysis of our text. We’re certain that your recommendations have substantially improved the overall quality of our research.

On the behalf of all Authors

Dr. Matteo RICCO’

Reviewer 2 Report

Overall: Very interesting topic and attempting to see if internet searches could be an epidemiologic surveillance tool for children is important.

Read through manuscript and ensure that all is in English as Italian is present mixed with English, for example Figure 3.

Abstract:

Would put a statement to explain the purpose was to see if online trends could be a surrogate marker for ILI. Please clarify.

Presented intercepts and p values for negative correlations but did not show correlations for Bronchitis. Would either include or exclude for all. Please clarify.

Bronchitis and Bronchiolitis should not be capitalized.

Introduction:

Appreciate the significance of paragraph 1 as RSV in children is very important, but as this manuscript discusses adults.  Need atleast a sentence discussing RSV in adults if data is being presented in this population. Lines 109-113 report the focus is on children also but then adults are placed in the manuscript. I would suggest either focusing on children and remove adult data or do children and adults and further enhance the introduction to include this population. Please add.

Lines 115-125 are methods and should be in the methods section.

Methods:

Ln 137, are these the symptoms published on the website or are these the symptoms the authors are choosing. If not, then I would consider the WHO IMCI characteristics for respiratory illness

Lines 86-89- One sentence. Should be added to one of the other paragraphs.

Lines 182-183: what does “In order to cope with the impact of non- 182 pharmaceutical interventions elicited by SARS-CoV-2 pandemic with the transmission of 183 respiratory pathogens including RSV [52–54], we assumed the reporting seasons 2017- 184 2020 as a reference, and calculated the estimates for excess incidence rates (EIR) in 2020- 185 2021 and 2021-2022 (i.e. the difference between the reported incidence rates [RIR] in a 186 given week in 2020-2021 and 2021-2022, and an estimate of the expected incidence rate 187 [EXR] for that time period) as follows” mean? Particularly in order to cope with the impact of the non-pharmaceutical interventions? This sentence does not make sense and is run on. Numerous studies have shown a lower incidence of respiratory viruses in children and presentations to healthcare centers during the COVID-19 pandemic, but this is not clear in this sentence if this was the goal----

Line 222- Is this a sensitivity analysis? Effector variables were affected by changes in other variables?

Lines 226-233: Multiple definitions used. Either explain defiitions for all pieces of methods or cite.

Results:

Figure 1. Axes are so small I can not see (particularly x axis). What are the dotted lines (vertical or smooth lines)? Legend does not make sense.

Figure 2- Increase the size of x axis . Running nose is misspelled. What are the dotted lines (vertical or smooth lines)?

I appreciate the correlation in Figure 6, but would like to see each data source overlayed to visualize lack of correlation.

Some of the results, particularly figures need to move to the Supplement.

Discussion:

First sentence mentions adults, but introduction does not. See comment above and please clarify.

Lines 541- What are non-pharmaceutical measures? Need to clarify incidences of respiratory disease in children declined. Well published. This needs to be referenced as this may greatly affect an unpowered study. This also gets to lines 550-554 which seem to present new data in the discussion. Would again consider how this data compares to other cohorts. Would look at Drakenstein cohort out of South Africa.

Other limitations – Use of broadband internet during pandemic, not wanting to go to hospital secondary to COVID and thus utilizing computer. This is started in lines 550-581. Some small limitations are noted but need to remark on methodology, population of Italy (access to internet, healthcare etc. )

Need to present how this compares to other studies.

Discussion feels like it is all limitations and needs to remark on comparison to other studies and other disease survelliance techniques. This will highlight if this is the correct approach and how maybe changes in methodology may create better correlations.

Conclusion:

I am not sure what it means. The last 2 sentences do not make sense and say the InfluNet is not accurate, but then we don’t know what is accurate based on this manuscript.

Author Response

Rev.2

Estimated Rev. 2,

First of all, thank you for your valuable comments, that have (we’re certain of it) substantially improved the overall quality of our article. Please see the following table in order to follow our interventions following your original recommendations.

Again, thank you for your considerable efforts:

Appreciate the significance of paragraph 1 as RSV in children is very important, but as this manuscript discusses adults.  Need at least a sentence discussing RSV in adults if data is being presented in this population. Lines 109-113 report the focus is on children also but then adults are placed in the manuscript. I would suggest either focusing on children and remove adult data or do children and adults and further enhance the introduction to include this population. Please add.

We agree with your recommendations. In fact, as our paper has been submitted to a journal focusing on Pediatric Age Patients and Diseases, despite the potential interest of some analyses on older age groups, we opted in for removing analysis and data on age groups older than 14 y.o.

Lines 115-125 are methods and should be in the methods section.

Thank you, we moved accordingly.

Ln 137, are these the symptoms published on the website or are these the symptoms the authors are choosing. If not, then I would consider the WHO IMCI characteristics for respiratory illness

Thank you: in fact, we revised our results in order to implement the US CDC recommendations for general public on symptoms associated with RSV. We think that such approach would provide a more objective pattern of symptoms to be explored in terms of their association with RSV rates.

Lines 182-183: what does “In order to cope with the impact of non- 182 pharmaceutical interventions elicited by SARS-CoV-2 pandemic with the transmission of 183 respiratory pathogens including RSV [52–54], we assumed the reporting seasons 2017- 184 2020 as a reference, and calculated the estimates for excess incidence rates (EIR) in 2020- 185 2021 and 2021-2022 (i.e. the difference between the reported incidence rates [RIR] in a 186 given week in 2020-2021 and 2021-2022, and an estimate of the expected incidence rate 187 [EXR] for that time period) as follows” mean? Particularly in order to cope with the impact of the non-pharmaceutical interventions? This sentence does not make sense and is run on. Numerous studies have shown a lower incidence of respiratory viruses in children and presentations to healthcare centers during the COVID-19 pandemic, but this is not clear in this sentence if this was the goal----

Thank you, this section has been extensively rewritten as follows:

We initially performed a descriptive analysis of the surveillance data on ILI by calculating estimates for each reporting seasons. In order to underline and then track down the impact of lockdown and non-pharmaceutical interventions (NPI, that is inter-ventions that communities can take to help slow the spread of illnesses that aim to prevent and/or control the pathogen’s transmission in the community)[6,7] elicited by SARS-CoV-2 pandemic on the transmission of respiratory pathogens including RSV [53–55], we as-sumed the reporting seasons 2017-2020, where no preventive measures were actually taken, as a reference, and calculated the estimates for excess incidence rates (EIR) in 2020-2021 (i.e. the season that was most heavily affected by lockdown and NPI), and 2021-2022, when confinement measures and NPI were substantially lifted. EIR was de-fined as the difference between the reported incidence rates [RIR] in a given week in 2020-2021 and 2021-2022, and the estimate of the expected incidence rate [EXR], calculated as an average for the index week i in the assessed timeframe, for the time period 2017-2020 as follows:

EIR = (RIRi - EXRi, 2017-2020) / EXRi, 2017-2020

Where:

RIRi,a = reported rates in a given week i

EXRi,2017-2020 = average rates in the given week i for the time period 2017-2020

Comparisons on seasonal incidence rates were then performed by means of the Analysis of the Variance (ANOVA), with Dunnet’s post-hoc test that assumed reporting season 2017-2018 as the reference group.

Lines 226-233: Multiple definitions used. Either explain defiitions for all pieces of methods or cite.

Thank you, we reworkerd this section as follows:

Eventually, infodemiological variables were assessed in terms of their potential autocorrelation. Autocorrelation can be defined as the correlation of a certain factor with itself over time, suggesting the underlying cyclic pattern of reported data [32]. Potential autocorrelation was ascertained through the calculation of Durbin-Watson (DW) statistics. DW test is a statistic test used to detect the presence of autocorrelation in the residuals (prediction errors) from a regression analysis [56]. The DW test statistic or d always lies between 0 and 4. If the d is substantially less than 2, there is evidence of positive serial correlation, while values greater than 2 suggest that no autocorrelation.

References 32 and 56 were therefore included:

32.       Bragazzi, N.L. Infodemiology and Infoveillance of Multiple Sclerosis in Italy. Mult Scler Int 2013, 2013, 1–9, doi:10.1155/2013/924029.

56.       White, K.J. The Durbin-Watson Test for Autocorrelation in Nonlinear Models 370 THE REVIEW OF ECONOMICS AND STATISTICS THE DURBIN-WATSON TEST FOR AUTOCORRELATION IN NONLINEAR MODELS; 1992; Vol. 74;.

Lines 541- What are non-pharmaceutical measures? Need to clarify incidences of respiratory disease in children declined. Well published. This needs to be referenced as this may greatly affect an unpowered study. This also gets to lines 550-554 which seem to present new data in the discussion. Would again consider how this data compares to other cohorts. Would look at Drakenstein cohort out of South Africa.

Thank you, we included the following information on NPI:

NPI, that is interventions that communities can take to help slow the spread of illnesses that aim to prevent and/or control the pathogen’s transmission in the community)[6,7]

Drakenstein cohort was included among the references in the discussion of the results as ref 81.

Other limitations – Use of broadband internet during pandemic, not wanting to go to hospital secondary to COVID and thus utilizing computer. This is started in lines 550-581. Some small limitations are noted but need to remark on methodology, population of Italy (access to internet, healthcare etc. )

We agree and included in the main text the following section:

Third, the first stages of SARS-CoV-2 pandemic and the associated lockdowns were associated with a sustained increase in the use of internet services, particularly where broadband connections were available. Even though the overall status of internet con-nections is usually acknowledged as far from optimal, an extensive use of internet services was identified also in Italy [95,96]. As a consequence, we cannot rule out that the increased research volumes identified during the reporting seasons 2019-2020, particularly in its late weeks, and then 2020-2021 may have been affected by the unprecedent referral to the new media for retrieving the needed information, with further limits when comparing post-pandemic figures with pre-pandemic ones.

Moreover, some further considerations have to be made about figures and tables.

As Rev.2 could appreciate, not only figures were extensively reworked (e.g. Fig.1) but we opted for simplifying data reporting moving in tables some of the figures previously included (particularly Figure 5-6-7), and the section that was previously and improperly labeled “sensitivity analysis” has been now reported as Appendix B as a series of correlation matrices, whose significance is now reported across the discussion as a potential source of limit to the reliability of regression analysis results because of the repeated and extensive correlation of assessed variables.

In summary, we thank Rev.2 for their contribution to our original paper, and we hope that our efforts may guarantee the eventual acceptance of our study on Children.

Dr. Matteo Riccò on the behalf of all Authors

Round 2

Reviewer 2 Report

 appreciate all the responses to the comments and they fulfill my concerns. Manuscript is much improved and easier to read. Few comments below. Would recommend proof reading as there are incorrect spellings and capitalizations  such as Seasonal in line 603 should not be capitalized. Could consider shortening as the manuscript is quite long and loses a bit of the message with the length. Could do this by moving some explanations and definitions to the supplement. 

Abstract-

Would re-read as there appears to be some typographical errors.

Ln 30- “with for”, please correct grammar

Ln 41- “Low Respiratory

Introduction:

Lines 83-86 should not be a separate paragraph. One sentence should not stand alone

Methods:

Ln 182-184- need to make more clear the term NPI. Would put NPI in parentheses by itself then start a new sentence explaining what it is.

Results:

Would  consider moving some tables to the supplement as the results section is very heavy. Would consider just highlighting the key results in the text. I am getting a bit lost in the volume of numbers amidst the text and I am not sure what the key results are. 

Author Response

Estimated reviewer,

thank you for your positive assessment of our study. We specifically addressed the concerns you've raised, as follows:

  • Would recommend proof reading as there are incorrect spellings and capitalizations  such as Seasonal in line 603 should not be capitalized.

We double checked the paper in order to remove as much as possible typos and annoying errors; please excuse us for the "far from optimal" status of the proof we've previously shared with you. Still, when dealing with the specific item from line 603, please note that we're dealing with the term "Seasonal Influenza", therefore (at least at our knowledge) it could/should be capitalized.

  • Could consider shortening as the manuscript is quite long and loses a bit of the message with the length. Could do this by moving some explanations and definitions to the supplement. 

We agree with your recommendation. In fact, we've moved Figures 4-5 to the Annex B (now figures B2 and B3), and the results sections on Table 4 was simplified, with a more extensive referral to the table.

  • Would re-read as there appears to be some typographical errors. Ln 30- “with for”, Ln 41- “Low Respiratory please correct grammar: 

Thank you, we fixed accordingly.

  • Lines 83-86 should not be a separate paragraph. One sentence should not stand alone

Thank you, we fixed accordingly;

  • Ln 182-184- need to make more clear the term NPI. Would put NPI in parentheses by itself then start a new sentence explaining what it is.

We agree with your recommendation, and the section was rewritten as follows:

We initially performed a descriptive analysis of the surveillance data on ILI by calculating estimates for each reporting seasons. In order to underline and then track down the impact of SARS-CoV-2 pandemic on the transmission of respiratory pathogens including RSV [53–55], we assumed the reporting seasons 2017-2020, where no preventive measures were actually taken, as a reference, and calculated the estimates for excess incidence rates (EIR) in 2020-2021, i.e. the season that was most heavily affected by lockdown and non-pharmaceutical interventions (NPI), and 2021-2022, when confinement measures and NPI were substantially lifted. NPI have been defined as interventions that communities can take to help slow the spread of illnesses that aim to prevent and/or control the pathogen’s transmission in the community)[6,7]. For the aims of this study, EIR was defined as the difference between the reported incidence rates [RIR] in a given week in 2020-2021 and 2021-2022, and the estimate of the expected incidence rate [EXR], calculated as an average for the index week i in the assessed timeframe, for the time period 2017-2020 as follows:

  • Would  consider moving some tables to the supplement as the results section is very heavy. Would consider just highlighting the key results in the text. I am getting a bit lost in the volume of numbers amidst the text and I am not sure what the key results are. 

Again, we agreed with your recommendation. As previously pointed out, we've moved Figures 4-5 to the annex section and extensively simplified the section of the main text on results from Table 4.

Eventually, we respectefully thank the present reviewer for the intervention they made and suggested, whose role was instrumental in radically improve the quality of this study.

MR